# The effects of temporal cues, point-light displays, and faces on speech identification and listening effort

**Katrina Sewell[1], Violet A. Brown[2], Grace Farwell[1], Maya Rogers[1], Xingyi Zhang[1], Julia F. Strand[1] ***

**1** Department of Psychology, Carleton College, Northfield, MN, United States of America, **2** Department of Psychological & Brain Sciences, Washington University in St. Louis, St. Louis, MO, United States of America

* jstrand@carleton.edu

**Data Availability Statement:** Data, code, and materials are available at: https://osf.io/jwyza/.

**Funding:** The work was supported by Carleton College and National Institute on Deafness and

## Abstract

Among the most robust findings in speech research is that the presence of a talking face improves the intelligibility of spoken language. Talking faces supplement the auditory signal by providing fine phonetic cues based on the placement of the articulators, as well as temporal cues to when speech is occurring. In this study, we varied the amount of information contained in the visual signal, ranging from temporal information alone to a natural talking face. Participants were presented with spoken sentences in energetic or informational masking in four different visual conditions: audio-only, a modulating circle providing temporal cues to salient features of the speech, a digitally rendered point-light display showing lip movement, and a natural talking face. We assessed both sentence identification accuracy and self-reported listening effort. Audiovisual benefit for intelligibility was observed for the natural face in both informational and energetic masking, but the digitally rendered point-light display only provided benefit in energetic masking. Intelligibility for speech accompanied by the modulating circle did not differ from the audio-only conditions in either masker type. Thus, the temporal cues used here were insufficient to improve speech intelligibility in noise, but some types of digital point-light displays may contain enough phonetic detail to produce modest improvements in speech identification in noise.

## Introduction

When listening to spoken language, seeing the talkers' face facilitates identification accuracy relative to hearing alone. This *visual enhancement (or audiovisual benefit)* has been demonstrated for listeners with typical hearing [1,2], those with hearing loss [3,4], cochlear implant users [5], older adults [6], and children [7]; and has been shown with a variety of stimuli, including syllables [8], words [9,10], sentences [11], and passages [12]. The visual information provided by a talking face is clearly a rich source of information that can help overcome the challenges associated with acoustic interference such as background noise or reverberation. But what is driving this audiovisual benefit?

Other Communication Disorders via a grant to Julia Strand (R15-DC018114). The funders had no role in study design, data collection and analysis, decision to publish, or preparation of the manuscript.

**Competing interests:** The authors have declared that no competing interests exist.

One way in which the visual signal benefits intelligibility is by revealing the locations of the lips, teeth, tongue, and other articulators, thereby providing cues about which phonemes are being produced. To identify the relative contributions of each of these articulators to visual enhancement, researchers have created stimuli in which only particular articulators are shown to participants. The earliest study of this kind recorded speech stimuli from a talker wearing luminous lipstick and black makeup on the rest of the face and teeth so only the lips were visible [13]. More recently, researchers have used point-light displays, which involve gluing glowing dots to several landmarks on the talker's face and recording in the dark so only the dots are visible [14]. Both of these methods have demonstrated that the movements of the lips in particular improve word identification accuracy relative to audio-only speech. This suggests that even minimal information about lip movement may provide sufficient phonetic detail to enhance speech intelligibility in noise.

In addition to providing information about the particular phonemes that are present (i.e., *what* the talker is saying), the visual signal also provides temporal cues that indicate *when* it is being said. Indeed, the area of the opening between the lips is correlated with the amplitude of the speech [15], suggesting that the visual signal may also cue the listener's attention to salient features of the acoustic stream (i.e., onset, offset, and moments of high relative amplitude), even if the phonetic cues provided by the visual signal are minimal. For example, audiovisual benefit can be obtained from severely blurred faces that lack the phonetic cues necessary to be accurately lipread [16]. Similarly, rotated audiovisual stimuli provide some intelligibility benefits relative to speech presented with a static face, suggesting that listeners may benefit from timing cues even when linguistic coherence is disrupted [17]. In addition this work using degraded or rotated faces, there is evidence that abstract stimuli that do not resemble faces can also provide informative temporal cues: Bernstein and colleagues [18] demonstrated that dynamic ovals and rectangles that modulated with the amplitude of the speech improved performance on a two-alternative forced choice task in which participants had to indicate which interval contained speech [see also 19].

Although visually-presented temporal cues that lack fine phonetic detail can improve listeners' performance on speech *detection* tasks, evidence is mixed regarding whether they benefit speech *identification*. Indeed, some prior work has shown that the presence of a dynamic circle that modulates with the amplitude of speech does not improve identification of syllables [20], words [21], or sentences [21,22] relative to hearing alone, whereas other work has demonstrated modest improvements in intelligibility from certain types of temporal cues including [23,24]. Thus, there is limited evidence that abstract temporal cues alone can benefit intelligibility. However, all existing work on how temporal cues affect speech identification has generated the modulating visual cue from the *acoustic* signal. For example, in Strand et al. [21], the size and luminance of the dot was modulated by the amplitude envelope of the speech. In spoken language, however, the movement of articulators can precede the acoustic input by up to a few hundred milliseconds [25] and may therefore alert listeners that speech is about to happen. Thus, the visual signal may act as a pre-cue that enables listeners to direct their attention to the acoustic signal. In contrast, visual stimuli that are generated from the acoustic input can only alert that speech is currently happening. Given the known advantages of pre-cueing for directing attention [e.g., 26], it may be that temporal cues facilitate speech identification only if they follow the time course of natural visual speech signals.

## Current study

The current study has four aims. First, it attempts to replicate prior studies using point-light displays but using a digital technique in which we generate visual displays from recordings of

talking faces—a novel technique in the audiovisual speech literature. A challenge of previous work on point-light displays is that stimuli are difficult to create and standardize because the procedure typically requires gluing dots to the speakers' articulators. The method introduced here involves using software that extracts facial landmarks from existing videos and generates simplified visual stimuli from them. This approach is easy to implement, can generate reproducible point-light displays, and can be applied to any video of a talking face.

A second goal of the study is to assess whether temporal cues alone can lead to visual enhancement when they mimic the time course of natural speech. Thus, rather than generating the modulating circle from the acoustic signal—whereby the size of the dot depends on the amplitude of the speech—we generated the dot from visual features. To accomplish this, we calculated the maximum displacement of the lips in the point-light displays described above and generated a circle in which the diameter reflects that maximum displacement (see Fig 1). This modulating circle provides the listener with some temporal cues to salient features of the acoustic input, and may also provide cues to vocal effort and vowel intensity. Note, however, that the modulating circle does not provide a complete picture of all the temporal information provided by a talking face. For example, it gives cues to maximum vertical lip displacement, but temporal cues from the edges of the mouth are not retained. In addition to the point-light displays (which contain isolated kinematic information about lip movements) and the modulating circle (which contains incomplete kinematic information about oral aperture and vowel intensity information), we also included a natural talking face to enable us to assess how much benefit the other visual stimuli provide relative to standard audiovisual stimuli.

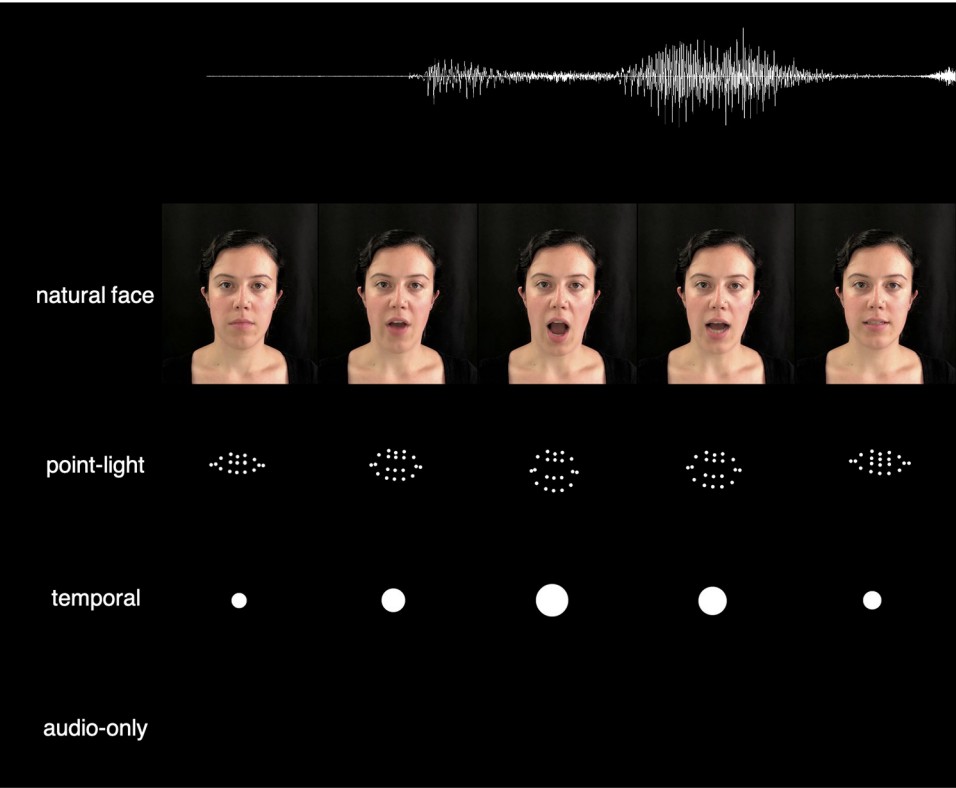

**Fig 1. Schematic showing the four conditions.** Note that the visual signals begin to change before the onset of the acoustic signal.

The third goal of the study is to assess whether these three types of visual stimuli have varying effects on speech identification accuracy across maskers (i.e., energetic vs. informational maskers). Energetic masking occurs when the frequencies present in the background noise overlap with those in the speech (thereby producing interference at a sensory level), and informational masking occurs when competing auditory signals draw the listener's attention away from the target speech (thereby producing interference at a cognitive level). Energetic maskers include steady-state and speech-modulated noise, and informational maskers include single- and multi-talker babble [note, however, that most informational maskers also produce energetic masking; see 27]. Some prior work suggests that natural faces may provide more release from masking when the masker is informational rather than energetic [9], but that finding has not been replicated, so this study will additionally test whether the magnitude of visual enhancement differs across masker types.

The final goal of this study is to assess the effects of various visual stimuli on listening effort —the cognitive resources necessary to understand speech [28,see 29]. Although audiovisual benefit is well established for speech identification, the literature is mixed regarding how visual speech cues affect listening effort [30]. The current study included a self-report measure of listening effort to evaluate whether and how minimal forms of visual input affect listeners' perception of the effort they expended to understand speech.

## Method

Stimuli, data, and code for analysis can be found at https://osf.io/jwyza/, and the preregistration documentation is available at https://osf.io/n4xys.

### Participants

We collected data from 152 participants to reach the preregistered sample size of 150 participants. Two participants were excluded because their mean word identification accuracy was more than three standard deviations below the mean in at least one of the visual conditions or noise types. The experiment was programmed and distributed using Gorilla Experiment Builder (Anwyl-Irvine et al., 2020), and participants were recruited via Prolific. Participants were paid $7.45 and the study took approximately 35 minutes to complete. The Carleton College Institutional Review Board approved all procedures. At the outset of the study, participants read an informed consent sheet and gave consent by checking a box. Enrollment was limited to Prolific users who had US IP addresses, listed their first language as English, reported normal or corrected-to-normal vision, had no hearing difficulties, and were 18–35 years old (M = 28.58, SD = 4.57). Participants self-reported the following demographics: 2% American Indian/Alaskan Native, 14.67% Asian, 77.33% White, 0.67% Native Hawaiian/other Pacific Islander, 8% African American; 63.33% male, 35.33% female; 16% Hispanic/Latino, 80% Not Hispanic/Latino (numbers may not add to 100% because participants were permitted to select multiple options or skip questions). Stimuli

**Speech stimuli.** Videos of the head and shoulders of a female talker without a discernible regional accent were obtained from Brown et al. [31]. Each sentence contained four keywords (e.g., "The *hot sun warmed* the *ground*"). Video stimuli for the point-light and temporal conditions were generated using the FacemarkLBF model [32], an existing facial landmark detection algorithm implemented in Python 3 using the cv2 module from the OpenCV library [33]. The program isolates individual frames of a video, identifies the location of the face within each frame, and then outputs the location of 68 facial landmarks that are stored as a vector of points (e.g., the outside of the right eye, the bottom middle of the noise), thereby transforming the videos into vectors of locations of facial features for each frame.

For the point-light condition, landmarks associated with required facial features (e.g., lips) were displayed as discrete white points on a black screen (see Fig 1). The stimuli created here included 20 dots, comparable to the number and location that have been used previously in physically-generated point-light displays [e.g., 14 dots; 14]. For the temporal condition, the modulating circle's radius was equal to the distance between the landmarks corresponding to the highest point of the top lip and lowest point of the bottom lip, reflecting the maximum displacement of the lips. The circle appeared onscreen when the lips began to move (i.e., the first frame in which the lip displacement changed from the onset of the video) and disappeared when the lips stopped moving. The audio recordings for all four stimulus types were identical for any given sentence. The visual display for the audio-only condition was a black screen.

**Maskers.** *Informational maskers.* Informational masking stimuli consisted of two-talker babble created by combining two recordings of different female speakers reading short, declarative sentences [e.g., "The clown had a funny face"; 34]. Recordings of the sentences produced by each talker were obtained from Van Engen [35], concatenated into a single continuous stream for each talker, matched on total RMS amplitude using Adobe Audition, and then combined into a single file. Given natural fluctuations in talking speed, sentences did not consistently start and end at the same time. We randomly sampled 150 different 4.5-second clips from a 7.5-minute stream of two-talker babble to generate the babble tracks for each trial, and randomly assigned these tracks to be yoked to target sentences. The combined babble tracks were presented with the speech at an SNR of -8.

*Energetic maskers.* Energetic masking noise consisted of speech-shaped noise that matched the long-term average spectrum of the speech stimuli. The speech-shaped noise was generated in Praat [36] and presented at an SNR of -8. SNRs for both masker conditions were determined via pilot testing to attempt to match audio-only identification accuracy in the two masking conditions.

**Procedure.** Stimulus presentation and data collection were conducted on participants' personal computers, and audio was presented diotically through personal headphones. Participants were asked to complete the study in a quiet space and wear headphones for the duration of the study. Before beginning the experiment, participants completed a browser sound check to ensure that the audio could be heard at a comfortable level, and then completed a headphone screening for web-based auditory experiments in which participants are presented with three 200-Hz sinusoidal tones and must identify the tone that sounds the quietest [see 37 for details]. Participants had a chance to redo the headphone screening if they failed the first time. Next, participants were presented with a sample sentence in silence to familiarize themselves with the target talker's voice, followed by the same sentence in white noise and then in babble. This familiarization process was then repeated with a second sample sentence. Participants then completed eight practice trials, one for each combination of visual condition and masker type.

During the main experiment, participants were presented with sentences one at a time and were instructed to type what they heard in a textbox after each sentence. Visual conditions (audio-only, temporal, point-light, and natural face) and masking noise (energetic, informational) were manipulated within-subjects. Conditions were blocked and counterbalanced, and each sentence appeared in each condition an equal number of times across participants, but each participant heard each sentence only once. Participants identified 19 sentences in each condition (76 keywords) for a total of 152 sentences (608 keywords) each.

In addition to the sentence identification task, participants completed the NASA Task Load Index (NASA-TLX) to assess self-reported listening effort twice per condition (i.e., after the first 10 sentences and at the end of each block). The NASA-TLX is widely used in research on listening effort and asks participants to evaluate mental demand, performance, effort, and

frustration using an unnumbered sliding scale (in this case ranging from 0–100). Data analysis was only performed on the effort subscale, but the other questions were included so that participants could differentiate performance on the task and the effort they exerted to attain that level of performance [38,39]. Following the main task, participants completed a demographic questionnaire asking about their age, race, biological sex, and native speaker status.

## Results

Data were analyzed using linear mixed effects models via the *lme4* package [40] in R [version 4.2.1; 41] and *p*-values for model parameters were obtained via the *lmerTest* package [42], which uses the Satterthwaite method for estimating degrees of freedom. Data were cleaned using the *tidyverse* suite of packages [43]. Statistical significance was assessed using likelihood ratio tests comparing models that differed only in the fixed effect of interest. We attempted to use the maximal random effects structure justified by the design for all analyses [44; see R script for details regarding the random effects structure of each model]. Below we first present the results separately for each noise type (energetic followed by informational masking), and then combine the data for the two noise types to assess whether audiovisual enhancement differs across maskers.

### Sentence identification accuracy

Given that each sentence contained four keywords, the outcome was entered into the model as a proportion of words correctly identified with weights indicating the denominator (i.e., grouped binomial data with weights = 4), with each row of the data frame corresponding to one sentence for one participant. In addition to counting completely accurate responses as correct, we also counted responses that were phonologically identical to target (e.g. "hi" to "high"), contained superfluous punctuation (e.g. "chance/" to "chance"), differed from target by one letter (provided that the input it not also a word; e.g., "answre" to "answer"), and otherwise correct responses that appeared to include a misplaced space (e.g., "th ecow" to "the cow"). Differences in pluralization between targets and responses (e.g., "cats" and "cat") were also counted as correct, but irregular differences in pluralization (e.g., "child" and "children") were not counted as correct. Corrections for pluralizations and homophones were done algorithmically (see R script) and decisions about other corrections were made by two independent coders blind to condition. Disagreements between those two coders were resolved by a third coder. The manual corrections accounted for 3% of participant responses. Participants and sentences were entered as random effects, and visual condition was entered as a fixed effect.

**Energetic masking.** Including visual condition in the model significantly improved model fit ($\chi^2_3$ = 213.2, *p* < .001, see Fig 2). Both the natural face (*B* = 1.35, *SE* = 0.07, *z* = 18.79, *p* < .001) and the point-light display (*B* = 0.23, *SE* = 0.05, *z* = 4.52, *p* < .001) improved identification accuracy relative to the audio-only condition, but the temporal and audio-only conditions did not differ from one another (*B* = 0.08, *SE* = 0.05, *z* = 1.70, *p* = .09). The point-light display improved identification accuracy relative to the temporal-only condition (*B* = -0.15, *SE* = 0.05, *z* = -2.86, *p* = .004), and the natural talking face led to significantly better identification than both the temporal-only condition (*B* = 1.27, *SE* = .07, *z* = 18.22, *p* < .001) and the point-light display (*B* = 1.12, *SE* = .07, *z* = 17.08, *p* < .001). Thus, the point-light display but not the modulating circle provided moderate audiovisual benefit, though substantially less than that provided by the natural face. The natural face versus temporal comparison was not of primary interest to any experiment in this study so we did not preregister those analyses. However, we opted to run these analyses so significance levels could be included in the figures for clarity.

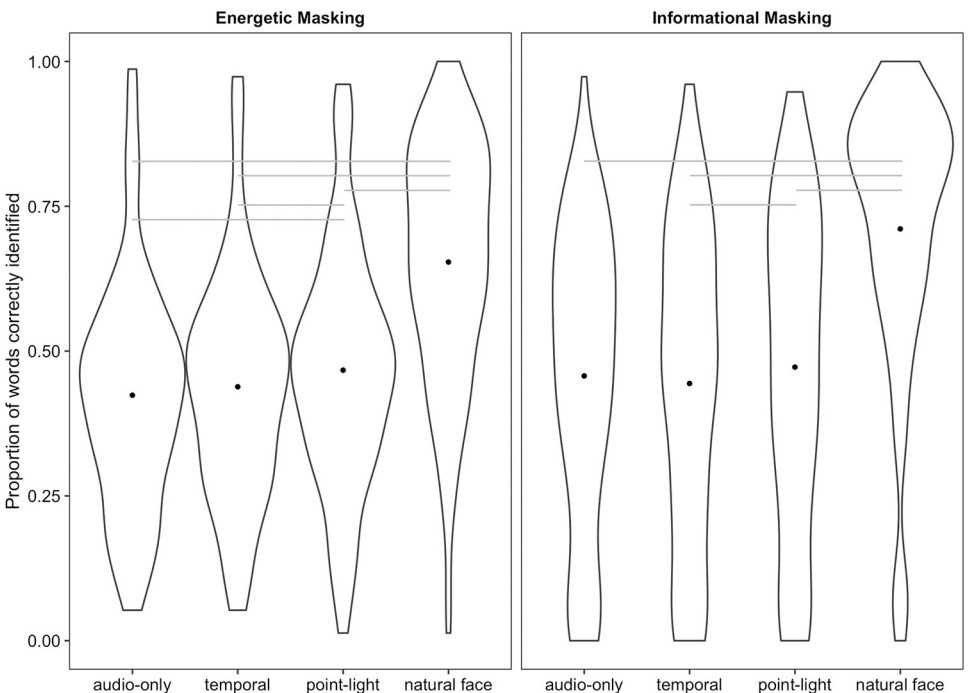

**Fig 2. Identification accuracy by visual stimulus type and masker condition.** Dots represent condition means. Gray lines represent statistically significant pairwise comparisons. A table with means and standard deviations for each condition appears in the Supplementary Materials.

**Informational masking.** A model that included a fixed effect for visual condition provided a better fit for the data than one that did not ($\chi^2_3 = 190.74$, $p < .001$). The natural talking face improved sentence identification accuracy relative to the audio-only condition ($B = 1.66$, $SE = 0.09$, $z = 17.49$, $p < .001$), but neither the modulating circle ($B = -0.11$, $SE = .07$, $z = -1.59$, $p = .11$) nor the point-light display ($B = 0.12$, $SE = 0.08$, $z = 1.41$, $p = .16$) led to visual enhancement. As in the energetic masking condition, accuracy in the point-light condition was significantly better than that in the temporal condition ($B = -.23$, $SE = .08$, $z = -2.79$, $p = .005$). Finally, the natural face led to significantly better performance than both the point-light display ($B = 1.54$, $SE = .10$, $z = 15.24$, $p < .001$) and the modulating circle ($B = 1.77$, $SE = .10$, $z = 16.96$, $p < .001$). Thus, the most notable difference between the pattern of results for information and energetic masking was that only the natural face provided visual enhancement relative in informational masking, whereas the point-light display additionally provided modest improvement in intelligibility in energetic masking.

**Comparing maskers.** Next, we subsetted the data to include only the audio-only and natural face conditions and combined the data from the energetic and informational maskers to assess whether masker type moderates the magnitude of audiovisual benefit. The full model contained fixed effects for visual stimulus, masker type, and the interaction. The reduced model was identical to the full model but omitted the interaction to assess whether audiovisual benefit differed across maskers. The full model provided a better fit for the data than the reduced model ($\chi^2_1 = 31.94$, $p < .001$), and the summary output for the full model also indicated a significant interaction ($B = -.29$, $SE = 0.05$, $z = -5.69$, $p < .001$). Participants showed moderately more audiovisual benefit with informational maskers (a 25.37% benefit) than with energetic ones (a 22.99% benefit; i.e., a difference in benefit between maskers of 2.38%), replicating previous work [9].

## Listening effort

**Energetic masking.**  Including visual condition in the model significantly improved model fit ($\chi^2_3$ = 28.82, $p < .001$; see Fig 3). Participants reported less subjective effort in the natural face condition than in all other conditions, including audio-only ($B$ = -7.66, $SE$ = 1.43, $t$ = -5.35, $p < .001$), temporal ($B$ = -7.19, $SE$ = 1.41, $t$ = -5.09, $p < .001$), and point-light ($B$ = -6.69, $SE$ = 1.28, $t$ = -5.23, $p < .001$). All other comparisons were non-significant: audio-only vs. temporal ($B$ = -.47, $SE$ = .79, $t$ = -.60, $p$ = .55), audio-only vs. point-light ($B$ = -.97, $SE$ = 0.91, $t$ = -1.06, $p$ = .29), and temporal vs. point-light ($B$ = 0.49, $SE$ = 0.83, $t$ = 0.60, $p$ = .55).

**Informational masking.**  A model that included a fixed effect for visual condition provided a better fit for the data than one that did not ($\chi^2_3$ = 60.20, $p < .001$). As in the energetic masker, participants reported less subjective effort in the natural face condition than in all other conditions, including audio-only ($B$ = -10.26, $SE$ = 1.21, $t$ = -8.49, $p < .001$), temporal ($B$ = -8.75, $SE$ = 1.34, $t$ = -6.54, $p < .001$), and point-light ($B$ = -8.09, $SE$ = 1.24, $t$ = -6.52, $p < .001$). However, unlike in the energetic masker, both the modulating circle ($B$ = -1.51, $SE$ = 0.71, $t$ = -2.11, $p$ = .04) and the point-light display ($B$ = -2.17, $SE$ = 0.77, $t$ = -2.83, $p$ = .005) led to lower effort ratings than the audio-only condition. The modulating circle and point-light displays did not differ from one another ($B$ = 0.66, $SE$ = 0.81, $t$ = 0.81, $p$ = .42).

**Comparing maskers.**  Finally, we assessed whether the reduction in subjective effort from adding a talking face was greater for informational than energetic masking. The full model contained fixed effects for visual condition (audio-only vs. face), masker type (energetic vs. informational), and their interaction. A model lacking the interaction term provided worse fit for the data than the model with the interaction ($\chi^2_1$ = 5.54, $p$ = .02), and the summary output for the full model also indicated a significant interaction ($B$ = 2.60, $SE$ = 1.11, $z$ = 2.36, $p$ = .02). The addition of a natural face decreased self-reported effort by 7.66 points (on a scale from

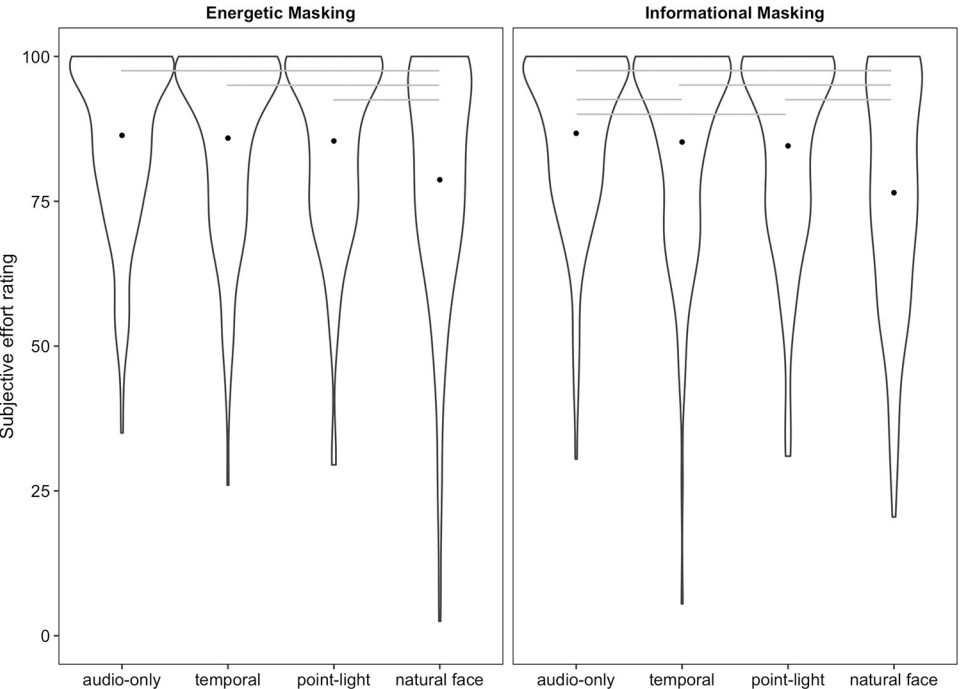

**Fig 3. Subjective effort rating by visual stimulus type and masker condition.** Dots represent condition means. Gray lines represent statistically significant pairwise comparisons. A table with means and standard deviations for each condition appears in the Supplementary Materials.

0–100) in the energetic masking condition, whereas the change was 10.27 points in the informational masker. Previous work has shown greater audiovisual benefit for informational masking than energetic masking in terms of identification accuracy [9], but this is the first study to demonstrate this effect for self-reported listening effort.

## Discussion

The first aim of this study was to test a new mode of generating point-light displays of talking faces. We found that the digitally-rendered point-light displays of the talker's mouth used here provided modest intelligibility gains relative to audio-only speech, but only in energetic masking. A major advantage of the digitally rendered point light displays is that they overcome some of the logistical challenges of gluing dots to a speaker's articulators. However, a disadvantage is that the digital technique may provide less precise information about the talker's facial features. For example, previous work using point light displays glued physical dots to the talker's mouth and face [14], which may have provided information in three dimensions. That is, because the dots were positioned at the outer and inner edges of the lips, bilabial consonants may have caused the dots on the inner edge to flatten, which would provide information about lip orientation as well as location. Future work that seeks to create digital point-light displays should therefore consider alternate methods that retain more detailed information about the relative positions of the articulators. Relatedly, given previous research indicating that the number and placement of the luminous points on the face affects the amount of audiovisual benefit derived from point-light displays [14], future researchers might also vary these parameters to assess whether this is also the case for digitally-rendered displays. Finally, it may be informative to compare multiple methods for generating digital point light displays to help clarify which facial landmarks or methods for extracting them provide the most audiovisual benefit.

The second aim of the study was to assess whether visual cues about the timing of the speech can benefit speech intelligibility in noise. We did not find any evidence that temporal information alone can lead to audiovisual benefit in either masker type, consistent with previous work [20–22]. Crucially, however, we used a novel technique in which we generated the visual timing cues from the *visual* signal rather than the *auditory* signal—thereby creating a visual signal that more closely approximates the timing of natural audiovisual speech—and still did not find intelligibility benefits. Taken together, these results provide converging evidence that temporal information alone is insufficient to lead to gains in speech intelligibility.

The third aim of the study was to evaluate whether the degree of intelligibility benefit provided by degraded and natural audiovisual stimuli differs across masker types. Many of the effects we observed here were consistent across the two maskers: Natural talking faces produced the best performance on the sentence identification task than any other condition, and the modulating circle providing temporal cues to speech did not improve intelligibility relative to audio-only speech. In contrast, point-light displays—which contain more phonetic cues than the modulating circle but far less than a natural talking face—only benefitted intelligibility in the energetic masker. It may be that the gross phonetic cues provided by the point-light displays help to compensate for some of the information that was lost in the degraded auditory signal, but do not provide sufficient detail to facilitate stream segregation and therefore only benefit intelligibility for maskers that primarily obscure phonetic information (i.e., energetic maskers). For example, they may contain enough detail to help the listener distinguish /pot/ from /got/, but not enough to enable reliable perceptual grouping between the digitally-rendered lips and the target speech stream. This may also explain why the intelligibility gains from point-light displays were quite small: These cues may only help to distinguish

phonologically similar words (i.e., neighbors) that differ in place of articulation, and only when the words contain phonemes that are visually distinct [45].

In a final analysis addressing differences in audiovisual benefit across masker types (aim three), we sought to replicate the finding that listeners gain greater benefit from natural audio-visual speech in informational than energetic masking [9]. We replicated this finding in our study, but it is worth noting that the effect size was quite small: Audiovisual benefit was only 2.38% larger in the informational than the energetic masker. Helfer and Freyman [9] did not report mean intelligibility values in each condition in their study, but visual inspection of Fig 1 in their paper suggests a larger effect in their work (particularly at the SNR of -4, which rendered intelligibility levels comparable to those observed here). Helfer and Freyman [9] propose that visual enhancement may be more pronounced in informational masking because the visual signal provides both phonetic supplementation *and* cues to stream segregation, and these additional attentional cues are less important in purely energetic maskers. It is unclear why the effect size observed in our study was smaller than that in the original study, but this may simply be driven by the fact that the degree of informational masking is highly dependent upon features of the particular targets and maskers used [e.g., more masking occurs when the target and masker are produced by talkers of the same sex; 46,47]. Regardless, our results provide converging evidence that listeners may gain slightly more benefit from the visual signal in informational relative to energetic maskers.

The final aim of the study was to assess whether multiple types of visual cues affect subjective listening effort. We found that in energetic masking, only natural faces reduced subjective effort relative to audio-only speech. In contrast, all three visual stimuli—modulating circles providing only temporal cues, point-light displays, and natural faces—reduced subjective effort in informational masking. These findings replicate prior work demonstrating that temporal cues reduce subjective effort in two-talker babble [21], and extend it to temporal displays that were derived from the visual (rather than auditory) signal and to digitally-generated point-light displays. Taken together, these results add to a growing body of work demonstrating that the effort required to understand speech is dissociable from the accuracy with which speech was identified; that is, manipulations that affect effort may not affect intelligibility [21,48,49], or may even have opposite effects on effort and intelligibility [30]. These findings underscore the importance of assessing both intelligibility and effort whenever possible. Indeed, in the context of our study, if we had only collected intelligibility data in the informational masker, we would have concluded that the minimal visual cues we included in our study (i.e., temporal-only cues and point-light displays) did not contain sufficient detail to affect performance on speech perception tasks. However, the subjective effort data suggest that although they may not affect identification accuracy, these reduced visual cues may affect other facets of the listener's experience.

## Supporting information

**S1 File. Pilot studies and additional descriptive statistics from the main experiment.** (PDF)

## Acknowledgments

We are grateful to Holly Griffiths and Michael Akeroyd for providing access to unpublished work, ZhaoBin Li for programming support, and Naseem Dillman-Hasso and Jed Villanueva for helpful discussions.

## Author Contributions

**Conceptualization:** Katrina Sewell, Violet A. Brown, Grace Farwell, Maya Rogers, Xingyi Zhang, Julia F. Strand.

**Data curation:** Violet A. Brown, Julia F. Strand.

**Formal analysis:** Violet A. Brown, Julia F. Strand.

**Funding acquisition:** Julia F. Strand.

**Investigation:** Violet A. Brown.

**Methodology:** Violet A. Brown, Grace Farwell, Maya Rogers, Julia F. Strand.

**Project administration:** Katrina Sewell.

**Resources:** Julia F. Strand.

**Software:** Maya Rogers, Xingyi Zhang.

**Supervision:** Julia F. Strand.

**Validation:** Violet A. Brown.

**Visualization:** Violet A. Brown, Julia F. Strand.

**Writing – original draft:** Violet A. Brown, Julia F. Strand.

**Writing – review & editing:** Katrina Sewell, Violet A. Brown, Grace Farwell, Maya Rogers, Xingyi Zhang, Julia F. Strand.

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
