## [Decision Letter · Decision Letter 0]

1 Aug 2023

PONE-D-23-05519The Effects of Temporal Cues, Point-Light Displays, and Faces on Speech Identification and Listening EffortPLOS ONE

Dear Dr. Strand,

Thank you for submitting your manuscript to PLOS ONE. After careful consideration, we feel that it has merit but does not fully meet PLOS ONE’s publication criteria as it currently stands. Therefore, we invite you to submit a revised version of the manuscript that addresses the points raised during the review process. Both reviewers found the study and manuscript to be clearly motivated, technically sound, and well-written. However, they had a few minor concerns that should each be addressed in the revision.

We look forward to receiving your revised manuscript.

Kind regards,

Andrew R Dykstra

Academic Editor

PLOS ONE

Journal Requirements:

2. Please provide additional details regarding ethical approval in the body of your manuscript. In the Methods section, please ensure that you have specified the name of the IRB/ethics committee that approved your study

3. Please provide additional details regarding participant consent. In the Methods section, please ensure that you have specified (1) whether consent was informed and (2) what type you obtained (for instance, written or verbal). If your study included minors, state whether you obtained consent from parents or guardians. If the need for consent was waived by the ethics committee, please include this information."

"The work was supported by Carleton College and National Institute on

Deafness and Other Communication Disorders via a grant to Julia Strand (R15-DC018114). Correspondence should be addressed to Julia Strand (jstrand@carleton.edu)."

"The work was supported by Carleton College and National Institute on Deafness and Other Communication Disorders via a grant to Julia Strand (R15-DC018114). The funders had no role in study design, data collection and analysis, decision to publish, or preparation of the manuscript."

5. We note that Figure [xxxx] includes an image of a [patient / participant / in the study].

7. Please remove your figures from within your manuscript file, leaving only the individual TIFF/EPS image files, uploaded separately. These will be automatically included in the reviewers’ PDF.

Reviewers' comments:

Reviewer's Responses to Questions

**Comments to the Author**

1. Is the manuscript technically sound, and do the data support the conclusions?

Reviewer #1: Partly

Reviewer #2: Yes

2. Has the statistical analysis been performed appropriately and rigorously? 

Reviewer #1: Yes

Reviewer #2: Yes

3. Have the authors made all data underlying the findings in their manuscript fully available?

Reviewer #1: Yes

Reviewer #2: Yes

4. Is the manuscript presented in an intelligible fashion and written in standard English?

Reviewer #1: Yes

Reviewer #2: Yes

5. Review Comments to the Author

Reviewer #1: Summary

This study tested the degree to which different types of visual information (full face, point-light lips, and a dot that modulates size) impact audiovisual speech identification and listening effort when the auditory signal is partially masked by energetic or informational noise. The results suggest that a full face provides the most visual benefit to audiovisual speech identification, with a point-light display of the talker’s lips providing a benefit in energic masking but not informational masking. In contrast, the modulating dot provided no visual benefit in either masking condition. Compared to the auditory only condition, visibility of the full-face reduced listening effort across masking conditions. Point light lips and the dot display did not affect listening effort in the energetic masking condition, but both reduced effort in the informational masking condition, somewhat in contrast with identification performance.

General Comments

The manuscript is well written, presenting clear motivation and hypotheses for the roles of different types of visual information and auditory masking that can affect audiovisual benefit and listening effort. The methods are largely sound, the results are clearly presented, and the interpretation of the results is mostly reasonable. I have only a couple points of concern that the authors should consider pertaining to their description of their conditions and the interpretation of their results that should be addressed before publication.

The authors describe their dot display as preserving temporal information. However, it can be argued that this display does not preserve all temporal information. Modulation of the dot’s size is determined by the time-varying oral aperture from the full face display, but aperture alone does not necessarily preserve all salient time-varying information available from a speakers articulating face. Additionally, and as the authors state in the introduction, oral aperture can relate to vocal effort and amplitude, meaning that the time-varying size of the dot can carry more than just temporal information. The authors should be clear about what information may be available from their dot display, and how information varies across their visual conditions (e.g., full face: all available visible information; point-light: isolated kinematic information for lip movements; dot display: non-kinematic information for oral aperture and vowel intensity information).

Additionally, the point-light display used in this study isolated points to the speaker’s lips, which can provide kinematic information for lip movements (intrinsically linked to lip aperture). However, previous studies have found that the number and placement of dots (i.e., lips, teeth, tongue, and surrounding areas) can affect the amount of kinematic information available from a point-light display and its effect on audiovisual speech perception (e.g., Rosenblum et al., 1996). I’m sure the software used to create the point-light displays for this study limits the placement and saliency of points derived from an articulating face, but the authors should consider how varying the number of points may affect their results.

References

Rosenblum, L. D., Johnson, J. A., & Saldaña, H. M. (1996). Point-light facial displays enhance comprehension of speech in noise. Journal of Speech and Hearing Research, 39(6), 1159-1170.

Reviewer #2: I have no major issues with the paper. The work seems carefully done, the data properly analyzed, and the paper clearly written. I have just a few comments that I hope will improve the manuscript before publication.

First, a quick request for adding line numbers in the future (for journals that do not auto-add them on submission, since two sets of line numbers is worse than none).

In the intro I am missing a few references to to two of Yi Yuan's studies:

https://pubs.aip.org/asa/jasa/article/147/3/EL246/997260/Visual-analog-of-the-acoustic-amplitude-envelope

https://pubs.asha.org/doi/abs/10.1044/2021_JSLHR-20-00688

It also might make sense to include Fiscella et al 2022 (https://link.springer.com/article/10.3758/s13414-022-02440-3), which compares AV benefit of a visual face to an upside down visual face (such that timing is preserved but linguistic information is partially disrupted).

Page 8, informational maskers: was each babble stream presented at -8, or was the sum of the streams at -8 dB SNR? In the former case, the total SNR would be about -11 dB.

Page 8, second line of Procedures: binaurally  diotically, unless there were binaural spatial cues

Page 9-10: the first two paragraphs of your results feel to me like they belong in the methods section.

Page 15, first paragraph: It may also just have some additional 2D noise. Did the authors visually inspect the dots overlaid on talker videos both in video and individual frames? There are *many* models for generating these landmarks, some better than others, and I am not familiar with the one used here. The dlib default, for instance, is quite bad in my experience. This one (https://twitter.com/alexcarliera/status/1678720831122186240) seems to be really excellent (I think that may be the first time I've referenced a tweet in a review). I am not sure if it is available for use or not, but it's an example of what's possible.

Page 16, second paragraph: The connection between the results of Helfer and Freyman and this study would depend on the slope of the psychometric function, correct? (which is not the subject of either study).

In their fig. 1 they show percent correct for three SNRs, one of which is -8 dB (used also in your study). To my eye, the gap between F-F A and F-F AV PC seem to differ by about 4% between the two plots.

While I agree your effect is small, I am a little uncomfortable with saying it is much smaller than the Helfer and Freyman effect, which also seems small from their figures and is reported in different units (unless I'm misunderstanding their results, which is entirely possible!).

6. PLOS authors have the option to publish the peer review history of their article (what does this mean?). If published, this will include your full peer review and any attached files.

Reviewer #1: No

Reviewer #2: No

---

## [Author Response · Author response to Decision Letter 0]

4 Aug 2023

We thank the editor and the reviewers for the time and energy that went into reviewing this paper! We’ve addressed the issues raised in the reviews and summarize the changes below (our responses here are indicated with asterisks). 

Journal Requirements:

*We have addressed all the journal requirements. The individual in figure 1 is the second author of the paper. The consent form seems intended for use by participants rather than by authors (e.g., it says “ have discussed this consent form with ___, who is an author of this article”). Therefore, it doesn’t seem appropriate to have her sign the consent form or reference it in the text. 

Reviewers' comments:

Reviewer #1: Summary

This study tested the degree to which different types of visual information (full face, point-light lips, and a dot that modulates size) impact audiovisual speech identification and listening effort when the auditory signal is partially masked by energetic or informational noise. The results suggest that a full face provides the most visual benefit to audiovisual speech identification, with a point-light display of the talker’s lips providing a benefit in energic masking but not informational masking. In contrast, the modulating dot provided no visual benefit in either masking condition. Compared to the auditory only condition, visibility of the full-face reduced listening effort across masking conditions. Point light lips and the dot display did not affect listening effort in the energetic masking condition, but both reduced effort in the informational masking condition, somewhat in contrast with identification performance.

General Comments

The manuscript is well written, presenting clear motivation and hypotheses for the roles of different types of visual information and auditory masking that can affect audiovisual benefit and listening effort. The methods are largely sound, the results are clearly presented, and the interpretation of the results is mostly reasonable. I have only a couple points of concern that the authors should consider pertaining to their description of their conditions and the interpretation of their results that should be addressed before publication.

The authors describe their dot display as preserving temporal information. However, it can be argued that this display does not preserve all temporal information. Modulation of the dot’s size is determined by the time-varying oral aperture from the full face display, but aperture alone does not necessarily preserve all salient time-varying information available from a speakers articulating face. Additionally, and as the authors state in the introduction, oral aperture can relate to vocal effort and amplitude, meaning that the time-varying size of the dot can carry more than just temporal information. The authors should be clear about what information may be available from their dot display, and how information varies across their visual conditions (e.g., full face: all available visible information; point-light: isolated kinematic information for lip movements; dot display: non-kinematic information for oral aperture and vowel intensity information).

*Great points, thank you! We’ve added text regarding this in the “current study” section.

Additionally, the point-light display used in this study isolated points to the speaker’s lips, which can provide kinematic information for lip movements (intrinsically linked to lip aperture). However, previous studies have found that the number and placement of dots (i.e., lips, teeth, tongue, and surrounding areas) can affect the amount of kinematic information available from a point-light display and its effect on audiovisual speech perception (e.g., Rosenblum et al., 1996). I’m sure the software used to create the point-light displays for this study limits the placement and saliency of points derived from an articulating face, but the authors should consider how varying the number of points may affect their results.

*Another good point. We’ve added a comment about that in the discussion

Reviewer #2: 

I have no major issues with the paper. The work seems carefully done, the data properly analyzed, and the paper clearly written. I have just a few comments that I hope will improve the manuscript before publication.

*Thank you so much for these kind words!

First, a quick request for adding line numbers in the future (for journals that do not auto-add them on submission, since two sets of line numbers is worse than none).

*This is a great point, thanks for the suggestion!

In the intro I am missing a few references to to two of Yi Yuan's studies:

https://pubs.aip.org/asa/jasa/article/147/3/EL246/997260/Visual-analog-of-the-acoustic-amplitude-envelope

https://pubs.asha.org/doi/abs/10.1044/2021_JSLHR-20-00688

*We’ve incorporated these citations. Thanks!

It also might make sense to include Fiscella et al 2022 (https://link.springer.com/article/10.3758/s13414-022-02440-3), which compares AV benefit of a visual face to an upside down visual face (such that timing is preserved but linguistic information is partially disrupted).

*Given the relatively large body of work assessing the extent to which variations in the signal quality of the talking face affect audiovisual benefit (e.g., faces with varying degrees of visual blur, upside down faces, small faces, etc.), and our desire to keep this paper short, we have opted not to include this or other similar citations in this paper.

Page 8, informational maskers: was each babble stream presented at -8, or was the sum of the streams at -8 dB SNR? In the former case, the total SNR would be about -11 dB.

*Good catch! The sum of the streams was presented at -8 SNR. We’ve made this change.

Page 8, second line of Procedures: binaurally  diotically, unless there were binaural spatial cues

*Done!

Page 9-10: the first two paragraphs of your results feel to me like they belong in the methods section.

*We prefer to keep the information about analyses and data cleaning in the Results section, near where the outcome of those analyses are described.

Page 15, first paragraph: It may also just have some additional 2D noise. Did the authors visually inspect the dots overlaid on talker videos both in video and individual frames? There are *many* models for generating these landmarks, some better than others, and I am not familiar with the one used here. The dlib default, for instance, is quite bad in my experience. This one (https://twitter.com/alexcarliera/status/1678720831122186240) seems to be really excellent (I think that may be the first time I've referenced a tweet in a review). I am not sure if it is available for use or not, but it's an example of what's possible.

*HA! We appreciate the Tweet (sorry, the X?). We did look at the videos and individual frames, but without another method for generating landmarks to compare it to, it’s difficult to know how accurately the point light displays are representing the facial movements. We added a sentence about this in the Discussion section and suggest that comparing multiple methods for generating landmarks might be informative, for exactly the reason you suggest.

Page 16, second paragraph: The connection between the results of Helfer and Freyman and this study would depend on the slope of the psychometric function, correct? (which is not the subject of either study). In their fig. 1 they show percent correct for three SNRs, one of which is -8 dB (used also in your study). To my eye, the gap between F-F A and F-F AV PC seem to differ by about 4% between the two plots. While I agree your effect is small, I am a little uncomfortable with saying it is much smaller than the Helfer and Freyman effect, which also seems small from their figures and is reported in different units (unless I'm misunderstanding their results, which is entirely possible!).

*It’s certainly true that the effect was relatively small looking at an SNR of -8, but the AV benefit is much larger at an SNR of -4, which rendered intelligibility levels comparable to those observed in this study. However, we’ve toned down the language to reflect the fact that our findings weren’t much smaller than those observed by Helfer and Freyman (2005).

---

## [Editor Report · Decision Letter 1]

9 Aug 2023

PONE-D-23-05519R1The Effects of Temporal Cues, Point-Light Displays, and Faces on Speech Identification and Listening EffortPLOS ONE

Dear Dr. Strand,

Thank you for submitting your manuscript to PLOS ONE. After careful consideration, we feel that it has merit but does not fully meet PLOS ONE’s publication criteria as it currently stands. Therefore, we invite you to submit a revised version of the manuscript that addresses the points raised during the review process. Please consider making the small changes suggested in the Additional Editor Comments. Please submit your revised manuscript by Sep 23 2023 11:59PM. If you will need more time than this to complete your revisions, please reply to this message or contact the journal office at plosone@plos.org. Please include the following items when submitting your revised manuscript:A rebuttal letter that responds to each point raised by the academic editor and reviewer(s). You should upload this letter as a separate file labeled 'Response to Reviewers'.A marked-up copy of your manuscript that highlights changes made to the original version. You should upload this as a separate file labeled 'Revised Manuscript with Track Changes'.An unmarked version of your revised paper without tracked changes. You should upload this as a separate file labeled 'Manuscript'.If applicable, we recommend that you deposit your laboratory protocols in protocols.io to enhance the reproducibility of your results. Protocols.io assigns your protocol its own identifier (DOI) so that it can be cited independently in the future. For instructions see: https://journals.plos.org/plosone/s/submission-guidelines#loc-laboratory-protocols. Additionally, PLOS ONE offers an option for publishing peer-reviewed Lab Protocol articles, which describe protocols hosted on protocols.io. Read more information on sharing protocols at https://plos.org/protocols?utm_medium=editorial-email&utm_source=authorletters&utm_campaign=protocols.

We look forward to receiving your revised manuscript.

Kind regards,

Andrew R Dykstra

Academic Editor

PLOS ONE

Journal Requirements:

**Additional Editor Comments:**

Dear Dr. Strand,

Thanks for addressing all the reviewer comments.

I'm not trying to drag this out any longer - again, apologies for how long it took to find reviewers! - but I do agree with R2 that Friscella et al (2022) and other, similar literature seems relevant. It needn't be long. A sentence or two comparing your work to theirs (and other similar) would suffice.

One other suggestions that's completely up to you: Maybe consider adding a tiny bit more interpretation at the end of the abstract?

Kindly, Andy

---

## [Author Response · Author response to Decision Letter 1]

15 Aug 2023

I’m not trying to drag this out any longer - again, apologies for how long it took to find reviewers! - but I do agree with R2 that Friscella et al (2022) and other, similar literature seems relevant. It needn't be long. A sentence or two comparing your work to theirs (and other similar) would suffice.

>>We added a couple of sentences citing the Fiscella paper and another related one (using blurred rather than inverted faces) to better contextualize our paper!

One other suggestions that's completely up to you: Maybe consider adding a tiny bit more interpretation at the end of the abstract?

>>Done!

---

## [Editor Report · Decision Letter 2]

17 Aug 2023

The Effects of Temporal Cues, Point-Light Displays, and Faces on Speech Identification and Listening Effort

PONE-D-23-05519R2

Dear Dr. Strand,

We’re pleased to inform you that your manuscript has been judged scientifically suitable for publication and will be formally accepted for publication once it meets all outstanding technical requirements.

Kind regards,

Andrew R Dykstra

Academic Editor

PLOS ONE
---

## [Editor Report · Acceptance letter]

10 Oct 2023

PONE-D-23-05519R2 

The Effects of Temporal Cues, Point-Light Displays, and Faces on Speech Identification and Listening Effort 

Dear Dr. Strand:

I'm pleased to inform you that your manuscript has been deemed suitable for publication in PLOS ONE. Congratulations! Your manuscript is now with our production department. 

Kind regards, 

on behalf of

Dr. Andrew R Dykstra 

Academic Editor

PLOS ONE